Comment on: Dynamics of the Askja caldera July 2014 landslide, Iceland, from seismic signal analysis: precursor, motion and aftermath

Tómas Jóhannesson, Jón Kristinn Helgason and Sigríður Sif Gylfadóttir

Icelandic Meteorological Office, Reykjavík, Iceland

Correspondence to: T. Jóhannesson, tj@vedur.is

Anne Schöpa et al. (2018) report on an analysis of seismic signals released by the Askja July 2014 rockslide, Central Iceland, and conclude from their analysis that the volume of material displaced by the slide was 35–80 million m³ and that the centre of mass was displaced horizontally by 1260±250 m and vertically by 430±300 m. Referring to Gylfadóttir et al. (2017) as source, they state that the volume of the slide was 12–50 million m³ according to geodetic surveys. We note that the volume of the rockslide according to Gylfadóttir et al. (2017) was actually reported as 20 million m³ based on field measurements that include multi-beam surveys of the bottom of Lake Askja, measurements of the lake level, and photogrammetric DEMs of the rockslide area on land, from before and after the slide. Larger, preliminary estimates for the volume of the rockslide (30–50 million m³, Helgason et al., 2014; >12 million m³ for the tongue that entered Lake Askja, Höskuldsson et al., 2015; 20–50 million m³, Sæmundsson et al., 2015; and 15–30 million m³, Gylfadóttir et al., 2016) were given in a memo with preliminary results and in conference presentations. They were based on initial estimates of the rise of the water level in the lake due to the slide and inaccurate estimates of the rockslide volume on land. These were revised after the processing of multi-beam and lake level data and photogrammetric DEMs was completed (Grímsdóttir et al., 2016), and the revised results were used in the tsunami modelling of Gylfadóttir et al. (2017). Some of these earlier reports and conference presentations are also referenced by Schöpa et al.

Our estimate for the volume of material mobilized by the rockslide is based on a reconstruction of the geometry of the sliding plane from the available field evidence. It is, however, hard to exclude the possibility that some rotational movement took place at a deeper level, leaving little evidence in the surface geometry of the rockslide after the event. The volume of the debris tongue in the lake, approximately 10 million m³, is rather well determined. The average horizontal displacement of this mass on the bottom of the lake is ~2000 m. If the horizontal displacement of the centre of mass of the mobilized material was as great as estimated by Schöpa et al., most of the debris mass that terminated on land must have moved from the starting area down to the run-out zone near the shore of the lake, which has an area of ~330 thousand m². This area is estimated as the part of the run-out area farther than 600 m from the highest part of the source area of the rockslide since material that travelled a shorter distance does not contribute much to the horizontal displacement of the centre of mass. The volume and centre-of-mass displacement estimated by Schöpa et al. correspond to average debris thickness of 75–210 m in this part of the run-out zone of rockslide on land. This thickness is difficult to reconcile with the available field measurements, in particular the thickness values near the higher end of this range. The field measurements indicate more than an order of magnitude less thickness of the debris tongue in most of this area. In combination, the volume and centre-of-mass displacement estimated by Schöpa et al. are therefore inconsistent with our field observations.

Schöpa et al. (2018) also report an averaged maximum sliding velocity for the rockslide of 7±0.7 m/s at the shoreline when the rockslide enters the lake from their seismic analysis, which is much lower than the impact velocity $U_0 = 31$ m/s used in the tsunami simulations of Gylfadóttir et al. (2017). Gylfadóttir et al. estimated their impact velocity by calibrating the tsunami model against measurements of the run-up of the tsunami wave around the lake so their velocity estimate is also a indirect estimate based on modelling. Higher up the slope, Schöpa et al. estimate a slide velocity up to ca. 18 m/s well before the rockslide entered the lake (this can be derived from their Figure 5b). Both these velocities estimated from the seismic analysis are surprisingly low in the middle of the path of a large rockslide released from an elevation of hundreds of meters. This needs to be discussed in a paper suggesting such remarkably low velocities. Both because a much higher velocity at the shoreline had been inferred by the tsunami modelling of Gylfadóttir et al., but perhaps more importantly because it is unreasonable from a physical standpoint for a rockslide that will propagate 2100 m farther into the lake and deposit half of its volume beyond the shoreline. Schöpa et al. discuss the discrepancy between the velocity obtained from their analysis and the frontal velocity when the rockslide enters the lake estimated Gylfadóttir et al. They suggest three possible explanations for this, "(i) the

limited applicability of a constant mass assumption in the waveform inversion, (ii) the fact that the inversion gives the velocity of the total landslide mass, whereas the tsunami modelling is calculating the velocity of the front of the slide, and (iii) uncertainties in the volume of the material sliding into the lake used for the modelling." These explanations may all matter but do not properly reflect the possibility that there is something fundamentally or seriously wrong with the analysis that causes this discrepancy. In this context, the uncertainty of the velocity of ±0.7 m/s presented by Schöpa et al. seems remarkable. It is also difficult, if not impossible, to account for the observed run-up of tsunami waves in the lake with a slide velocity as low as deduced by Schöpa et al. from their seismic analysis.

We note that the rockslide may be expected to have been retarded somewhat at the shoreline from the maximum velocity farther up the path because it has propagated ~600 meters over relatively flat terrain from the location where it reached its highest velocity before it comes to the shoreline. Therefore, the velocity estimate of 31 m/s at the shoreline by Gylfadóttir et al. corresponds to considerably higher velocity higher up in the path where the analysis of Schöpa indicates a averaged maximum velocity of ca. 18 m/s. The maximum velocity of ca. 18 m/s, that may be inferred from Figure 5b in Schöpa et al., corresponds to the foot of the slope before the rockslide enters the run-out zone and where there presumably has been comparatively little retardation of the rockslide mass. The kinetic energy corresponding to this velocity is ~160 J/kg. For comparison, the average potential energy released in the descent of the rockslide mass down ca. 250 m vertically may be estimated on the order of 2500 J/kg. A velocity of ca. 18 m/s at the foot of the slope, therefore, implies that more than 90% of the original potential energy of the rockslide is already dissipated by friction when the rockslide enters the run-out zone and has 2700 m farther to go into the run-out zone and the lake. Of course the center of mass of the moving material does not propagate this far into the lake. It nevertheless seems dynamically implausible that the rockslide propagates at ca. 18 m/s at the foot of the slope after falling down ca. 250 m, as well as propagating at ca. 7 m/s at the shoreline with 2100 m farther to go. The estimated velocity of 7 m/s at the shoreline corresponds to the potential energy of 25 J/kg which is equivalent to the potential energy of an object raised vertically by only 2–3 m. We further note that both 7 m/s (in the middle of or somewhat above the middle of the run-out zone) and 18 m/s (maximum velocity at the start of the run-out zone) are quite low velocities for large rockslides with volumes of tens of millions of m³, see e.g. inferred velocities of several large langslides qouted by Evans et al. (2006).

There are few direct measurements of the velocity of large rockslides, and volumes are often uncertain because of difficulty locating the sliding surface. Interpretation of seismic data to estimate volumes and velocities of landslides is, therefore, interesting and could be useful in the context of hazard management, and, of course, for general understanding of rockslides as a geophysical phenomenon. The disagreement between the results of Schöpa et al. and available field measurements for the Askja 2014 rockslide indicates that further development is needed to obtain quantitative information about rockslide dynamics from the seismic signal analysis that they employ.

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
