# Peer review of "Comment on: Dynamics of the Askja caldera July 2014 landslide, Iceland, from seismic signal analysis: precursor, motion and aftermath"

_Earth Surface Dynamics, 2019_

## Referee Comment (RC1) · Tim Greenfield (Referee) · 17 Sep 2019

The authors comment on a recent (2018) paper by Schöpa et al. on the dynamics of the July 2014 landslide in Askja caldera, Iceland. Two criticisms of the paper are presented. The first is that the landslide volume as calculated by Schöpa et al. (2018) is too large compared to detailed field surveys and tsunami modelling presented in an earlier paper by Gylfadóttir et al. (2017). The second is that the maximum average velocity calculated by Schöpa et al. (2018) is an order of magnitude smaller than the results suggested by Gylfadóttir et al. (2017).

Schöpa et al. (2018) note in their text the many assumptions made in their modelling

procedure. They identify the length of the run-out as a key parameter in estimating the mass (and therefore volume) of the slide. Given the slide entered the lake and the difficulty in determining the basal slip plane (identified by the authors of this comment), there is significant uncertainty in this parameter. I note though, that Schöpa et al. (2018) do misquote the field and geodetic based estimates of volume from Gylfadóttir et al. (2017) in their text and these could have been used to estimate the run-out from the Schöpa et al. (2018) force time history. It is also interesting to note that another application of the Ekstrom & Stark (2013) technique in Greenland (Chao et al., 2019) overestimates the volume compared to accurate field and satellite-based surveys. This could suggest a weakness of the Ekstrom & Stark (2013) technique when a slide enters the water.

The velocity of the slide is quoted as an average maximum sliding velocity (7±0.7 m/s) and does, at first glance, appear to be slower than expected. This value follows directly from the estimation of the slide's mass and so low values may be expected given Schöpa et al. (2018) overestimate this parameter. Schöpa et al. (2018) note the discrepancy between their value and the 30 m/s estimated from tsunami modelling (Gylfadóttir et al., 2017). They highlight three simplifying assumptions in their modelling which may account for the difference between the two values. I believe however, that part of the misunderstanding comes from Schöpa et al. (2018) quoting an average velocity rather than a peak velocity. I calculate (from figure 5b in Schöpa et al., 2018) a peak velocity of 18 m/s (Figure 1). This is of the right order of magnitude and should be highlighted by the authors of this comment. It would be interesting to see whether the velocity-time history could produce the observed tsunami run-ups.

The text is fairly well written, although the second paragraph took me a couple of read throughs to really understand what the authors are trying to get across. I would suggest simplifying and clarifying the first two sentences, which are quite long.

I recommend publication of the comment after minor revisions to the text are completed.

**ESurfD**
[Figure]

**Fig. 1.** Velocity-time history from Schöpa et al. (2018). The digitised east, north and up components are shown by the thin lines and the overall speed by the thick red line.

---

## Author Comment (AC1) · 4 Oct 2019

We thank the reviewer (Tim Greenfield) for his comments. We will rephrase the second paragraph that he found difficult to read to make it more easily understandable.

Regarding our second criticism mentioned by the reviewer, that the maximum average velocity at the shoreline calculated by Schöpa et al. (2018) ($7\pm0.7$ m/s) is almost an order of magnitude smaller than suggested by the modelling of Gylfadóttir et al. (2017) (31 m/s), we note in our comment that this low magnitude of the velocity is also physically unreasonable in the middle of the path of a rockslide released from an elevation of hundreds of meters. This discrepancy needs to be discussed in a paper suggesting

such remarkably low velocity, of course because a much higher velocity at the shoreline had been obtained by tsunami modelling and published by Gylfadóttir et al. (2017), but perhaps more importantly because it is unreasonable from a physical standpoint for a rockslide that will propagate 2100 m farther into the lake and deposit half of its volume beyond this point. It is true that Schöpa et al. discuss the discrepancy between the velocity obtained from their analysis and the frontal velocity when the rockslide enters the lake estimated Gylfadóttir et al. They suggest three possible explanations for this, "(i) the limited applicability of a constant mass assumption in the waveform inversion, (ii) the fact that the inversion gives the velocity of the total landslide mass, whereas the tsunami modelling is calculating the velocity of the front of the slide, and (iii) uncertainties in the volume of the material sliding into the lake used for the modelling." These explanations may all matter but do not properly reflect the possibility that there is something fundamentally or seriously wrong with the analysis that causes this discrepancy. In this context, the uncertainty of the velocity of $\pm 0.7$ m/s presented by Schöpa et al. seems remarkable.

The reviewer notes that the seismic analysis of Schöpa leads to a maximum velocity of ca. 18 m/s somewhat higher up the slope before the slide approaches the shoreline, which is on the correct order of magnitude compared with the velocity estimate of 31 m/s by Gylfadóttir et al. (2017). We will in our revision clarify that both velocity estimates discussed in our comment apply to the point where rockslide enters the lake. We also note that the rockslide may be expected to have been retarded somewhat at the shoreline from the maximum velocity farther up the path because it has propagated $\sim$600 meters over relatively flat terrain from the location where it reached its highest velocity before it comes to the shoreline. Therefore, the velocity estimate of 31 m/s at the shoreline by Gylfadóttir et al. corresponds to considerably higher velocity higher up in the path when the analysis of Schöpa indicates that the velocity is at maximum ca. 18 m/s. The maximum velocity of ca. 18 m/s, that may be inferred from Fig. 5b in Schöpa et al., corresponds to the foot of the slope before the rockslide enters the run-out zone and where there presumably has been comparatively little retardation of the rockslide

mass. The kinetic energy corresponding to this velocity is ∼160 J/kg. For comparison, the average potential energy released in the descent of the rockslide mass down ca. 250 m vertically may be estimated on the order of 2500 J/kg. A velocity of ca. 18 m/s at the foot of the slope, therefore, implies that more than 90% of the original potential energy of the rockslide is already dissipated by friction when the rockslide enters the run-out zone and has 2700 m farther to go into the run-out zone and the lake. Of course the center of mass of the moving material does not propagate this far into the lake. It nevertheless seems dynamically implausible that the rockslide propagates at ca. 18 m/s at the foot of the slope after falling down 250 m, as well as propagating at ca. 7 m/s at the shoreline with 2100 m farther to go. We further note that both 7 m/s (in the middle or above the middle of the run-out zone) and 18 m/s (maximum velocity at the start of the run-out zone) are quite low velocities for large rockslides with volumes of tens of millions of m$^3$, see e.g. inferred velocities of several large langslides qouted by Evans et al. (2006).

Evans, S. G., Scarascia Mugnozza, G., Strom, A. L., Hermanns, R. L., Ischuk, A., and Vinnichenko, S.: Landslides from massive rock slope failure and associated phenomena. In: Evans, S. G., Scarascia Mugnozza, G., Strom, A. L., and Hermanns, R. L. (eds.), Landslides from Massive Rock Slope Failure. NATO Science series IV. Earth and Environmental Sciences, 49, Springer, 2006.

---

## Short Comment (SC1) · 14 Oct 2019

Anne Schöpa and Wei-An Chao

We thank the authors for their comment on our paper. Unfortunately, the comment was not submitted during the review process of our manuscript. We would like to take up the points mentioned by Jóhannesson et al. by explaining our inversion modelling and its results in the following paragraphs in detail.

We acknowledge the results of Gylfadóttir et al. (2017) that the horizontal displacement of the mobilised landslide mass at the bottom of the lake was about 2000 m with

a deposit volume of 10 million m$^3$. However, these parameters were not required for our inversion. The landslide seismic inversion adopted in Schöpa et al. (2018) was conducted with the assumptions (i) of a block model with time-independent landslide mass and (ii) that the long-period (LP: 12.5-50 seconds) seismic signals were mainly induced by landsliding along a planar failure surface on land. To satisfy the assumption of a constant block mass, Schöpa et al. (2018) only interpreted the landslide dynamics inferred from the LP seismic records when the mass was sliding on land. After a landslide mass enters a lake, it disintegrates and hence the seismic energy generated by a moving mass underwater is dissipated rapidly. Previous studies have demonstrated that seismic signals caused by sediment transport underwater exhibit relatively higher frequencies (> 1 Hz, Hsu et al., 2011; Chao et al., 2015). The seismic stations far away from the source could not capture these short-period signals induced by the movement of the submerged mass.

Landslide volume and trajectory: Once the landslide force-time history (LFH) is derived from the LP seismic waveform inversion, the acceleration of the center of the block mass can be computed by dividing the LFH by a constant mass. The displacement is found by a double integration of the acceleration. We used three frequency bands (0.02-0.05 Hz, 0.02-0.08 Hz, 0.04-0.08 Hz) for the source inversion (Schöpa et al., 2018). The computed LFH gave a sliding mass in the range of 7-16×10$^{10}$ kg by fitting the runout distance on land ( 1200 m, from the center of mass of the source area to the lakeshore, from satellite images and field observations). Assuming an average density of 2000 kg/m$^3$, 35-80 million m$^3$ of landslide volume was obtained. This value overestimated the landslide volume compared to the 20 million m$^3$ reported by Gylfadóttir et al. (2017). We attribute this discrepancy to (i) the underestimation of the runout distance used in the seismological determination of the landslide mass. In other words, the initially submerged sliding material may have contributed to the LP seismic signals; thus the mass derived from the trajectory needs to be updated. Longer runout path results in smaller mass of the sliding block. However, we note that seismic analysis can provide a constraint on the upper limit of a landslide mass. We further attribute
the discrepancy of the landslide volumes to (ii) the limitation of applying a constant mass assumption in the waveform inversion, (iii) the effects of a rotational block slide, and (iv) uncertainties in the volume computation resulted from the poor constraint of the sliding surface. Nevertheless, we apologise for unintentionally not having quoted Gylfadóttir et al. (2017)'s estimation of the landslide mass correctly.

Sliding velocity: Schöpa et al. (2018) stated the fact that the waveform inversion gives the spatially and temporally averaged velocity of the whole sliding block on land, whereas the tsunami modelling is adopting the velocity ($U_0$ in Gylfadóttir et al., 2017) of the front of the slide entering the lake. Therefore, a comparison between these two velocity values is difficult to make. In our work, we listed possible reasons for the discrepancy of the velocities (Schöpa et al., 2018). We have observed a late-arriving seismic phase (Fig. 5d in Schöpa et al., 2018) in the high-frequency envelope waveform recorded by the closest station, which might be induced by parts of the sliding material hitting the shoreline and moving into the lake. A similar observation of seismic signals has been reported by Chao et al. (2016). A possible solution would be to obtain the front velocity of the submerged mass by using the seismic radiation energy of the late-arriving signals.

We also noticed that the $U_0$ value derived from the tsunami modelling of Gylfadóttir et al. (2017) is very sensitive to the friction coefficient ($\mu$), which ranges from 0.15 to 0.30 for the majority of rockslide configurations. With the fixed input parameters, such as $\mu$, total deposit volume, drag coefficient ($C_d$), and add mass coefficient ($C_m$), U0 and the block thickness (d) are obtained through a grid-search scheme by fitting the observed water level of the lake. The reliability of this optimisation procedure is mainly controlled by the uncertainties in the fixed parameters. Before having a detailed comparison between seismologically-determined and tsunami-based impact velocities, sensitivity tests for these fixed parameters in the tsunami modelling are required.

Landslides occurring in coastal and lakeside regions can generate destructive tsunami waves when the mass slides into the water, which can pose a series of hazards to the
coastal or lakeside population. The volume of the sliding mass that enters the water is the crucial parameter for tsunami-wave simulations. Our recent study (Chao et al., 2018) showed how seismic techniques using real-time seismic records can provide estimates of essential physical parameters (i.e., sliding volume) of landslides, which can be utilised for near-real-time tsunami wave simulations. A combined analysis of the real-time seismic waveform inversion and of forward tsunami-wave modelling could enable timely operational warnings before the arrival of the destructive tsunami waves.

References:

Chao, W. A., Y. M. Wu, L. Zhao, V. C. Tsai and C. H. Chen (2015) Seismologically determined bedload flux during the typhoon season, Sci. Rep., 5, 8261; doi: 10.1038/srep08261.

Chao, W. A., L. Zhao, S. C., Chen, Y. M. Wu, C. H. Chen and H. H. Huang (2016) Seismology-based early identification of dam-formation landquake events, Sci. Rep., 5, 19259, doi: 10.1038/srep19259.

Chao, W. A., T. R. Wu, K. F. Ma, Y. T. Kuo, Y. M. Wu, L. Zhao, M. J. Chung, H. Wu and Y. L. Tsai (2018) The large Greenland landslide of 2017: Was a tsunami warning possible?, Seismol. Res. Lett., 89(4), 1335-1344.

Hsu, L., N. J. Finnegan and E. E. Brodsky (2011) A seismic signature of river bedload transport during storm events, Geophys. Res. Lett., 38, L13407

Gylfadóttir, S. S., J. Kim, J. K. Helgason, S. Brynjólfsson, Á. Höskuldsson, T. Jóhannesson, C. B. Harbitz and F. Løvholt, (2017) The 2014 Lake Askja rockslide-induced tsunami: Optimization of numerical tsunami model using observed data. J. Geophys. Res. Oceans, 122, 4110-4122, doi:10.1002/2016JC012496.

Schöpa, A., W. A. Chao , B. Lipovsky, N. Hovius, R. S. White, R. G. Green and J. M. Turowski (2018) Dynamics of the Askja Caldera July 2014 landslide from seismic signal analysis: precursor, motion and aftermath, Earth Surf. Dyn., 6, 467–485,

https://doi.org/10.5194/esurf-6-467-2018.

**ESurfD**

Interactive
comment

---

## Referee Comment (RC2) · Anonymous Referee #2 · 29 Nov 2019

Review of Esurf manuscript "Comment on: Dynamics of the Askja caldera July 2014 landslide, Iceland, from seismic signal analysis: precursor, motion and aftermath by Tómas Jóhannesson et al."

The comment-manuscript addresses several quantitative disagreements, for the same landslide, of the Schöpa et al. relatively to the Gylfadóttir et al. (2017) paper, respectively. It questions the strength of landslide parameters that are derived from seismic signal analysis (e.g. Schöpa et al. 2018).

First, from geometry patterns, the submitted "comment" point on (i) a false citation when reporting on the landslide volume as referenced by another study (i.e. Gylfadóttir et al. (2017). "Referring to Gylfadóttir et al. (2017) as the source, Schöpa et al. state that the volume of the slide was 12–50 million m according to geodetic surveys...whereas the referenced paper quantitatively points on a 20 million m value". Moreover, (ii) the volume and centre-of-mass displacement estimated by Schöpa et al. correspond to average debris thickness, which is more than an order of magnitude less thickness of the field measurements for the debris tongue Gylfadóttir et al. (2017).

Second from kinetic approaches, the submitted comment-manuscript points on the velocity disagreement between the 7 m/s sliding velocity from seismic signal analysis (Schöpa et al. 2018) and the 30 m/s impact velocity (as tuned to fit the observed tsunami run-ups around the lake by Gylfadóttir et al. 2017). All these points are well grounded when the comment authors should also explicitly state the Gylfadóttir et al. velocity estimate is an indirect measurement.

Before the comment to be accepted, I suggest some sentences should be added to the text for the comment-manuscript to go beyond the Gylfadóttir et al. 2017 versus Schöpa et al. binary analysis. e.g., What are the lessons to be learned for the landslide community from the discrepancies between the two studies? It may point on a necessity to switch from the deterministic outputs of both the Gylfadóttir et al. 2017 and Schöpa et al. 2018 studies to a more probabilistic approach where ensemble solutions are provided explicitly for geometry and kinematic of landslides.

specific comments: "A maximum velocity of only 7 m/s (corresponding to the potential energy of an object raised 50 vertically by 2–3 m) seems unreasonably low since this would imply a delicate local balance between frictional forces and the potential energy released at each instance during the fall, which does not seem likely" The above comment is qualitative in several parts. A more quantitative version is expected.

---

## Author Response (AR1)

Reykjavík, 27.1.2020

**Author's response regarding the revised version of the manuscript**

"Comment on: Dynamics of the Askja caldera July 2014 landslide, Iceland, from seismic signal analysis: precursor, motion and aftermath"

By Tómas Jóhannesson, Jón Kristinn Helgason and Sigríður Sif Gylfadóttir

Icelandic Meteorological Office, Reykjavík, Iceland

Correspondence to: T. Jóhannesson, tj@vedur.is

We thank Tim Greenfield and an anonymous reviewer for suggestions that improved our comment. We have rewritten the second paragraph that Tim Greenfield found hard to read and now include discussion of the higher velocity estimated by the seismic analysis of Schöpa higher up the path as suggested by him. We have expanded our discussion of the unrealistically low velocity with an added reference and we have omitted a sentence about which the anonymous reviewed commented "The above comment is qualitative in several parts" and replaced it with this expanded discussion. We also note that velocity estimates of both Gylfadóttir et al. and Schöpa et al. are "indirect measurement" as suggested by the anonymous reviewer.
The anonymous reviewer suggests that we add a discussion about "the lessons to be learned for the land-slide community from the discrepancies between the two studies? It may point on a necessity to switch from the deterministic outputs of both the Gylfadóttir et al. 2017 and Schöpa et al. 2018 studies to a more probabilistic approach where ensemble solutions are provided explicitly for geometry and kinematic of landslides". We intended to write only a short comment to draw attention to the problems that we find with the analysis in Schöpa et al. Expanding the comment to include discussion along this line would widen the scope of comment too much in our opinion so we did not elaborate on this point. We point out that the analysis of Gylfadóttir included model calibration that explored a large set of potential slide parameters and in that sense included some statistical aspects that are along the lines suggested by the reviewer.

Yours sincerely,
Tómas Jóhannesson